# Hotspot Information Network and Domain Knowledge Graph Aggregation in Heterogeneous Network for Literature Recommendation

**Wei Chen** [1,2,3,*] , **Yihao Zhang** [1] , **Yantuan Xian** [1,2] and **Yonghua Wen** [1,2]

[1] School of Information Engineering and Automation, Kunming University of Science and Technology, Kunming 650500, China
[2] Yunnan Key Laboratory of Artificial Intelligence, Kunming University of Science and Technology, Kunming 650500, China
[3] Yunnan Key Laboratory of Computer Technologies Application, Kunming University of Science and Technology, Kunming 650500, China
* Correspondence: chenwei1983@kust.edu.cn

**Abstract:** Tremendous academic articles face serious information overload problems while supporting literature searches. Finding a research article in a relevant domain that meets researchers' requirements is challenging. Hence, different paper recommendation models have been proposed to address this issue. However, these models lack a more comprehensive analysis of the connections between the literature, the domain knowledge provided, and the hotspot information expressed in the literature. Previous models make it impossible to locate the appropriate documents for domain literature. Additionally, these models encounter problems such as cold start papers and data sparsity. To overcome these problems, this paper presents a recommendation model termed PRHN. Inputs of the model are the hotspot information network and the domain knowledge graph, which both were developed during the preceding research phase. After the query terms are extracted and the associated heterogeneous literature networks are formed, they are aggregated in a uniform hidden space. Similarity with the candidate set is determined to transform the search problem into a TOP $N$ recommendation problem. Compared to state-of-the-art models, results generated by PRHN on public available datasets show improvement in HR and NDCG. Concretely, results on the metallurgical literature dataset are more conspicuous, with more remarkable improvement in HR and NGCC by approximately 4.5% and 4.2%.

**Keywords:** knowledge graph; hotspot information network; heterogeneous academic network; recommender system





## 1. Introduction

The rise of information technology and the Internet have made it difficult for users to access important information in the face of vast amounts of data. A range of data mining tools have been developed to address the issue of difficult access to important data in the era of information overload. A recommendation system is an essential technique for rapidly filtering and obtaining information from huge amounts of data, particularly in the challenges of personalized recommendation, changing input data, cold data start, and collaborative filtering of related data. Recommendation systems are widely utilized in web applications, ranging from search engines and e-commerce to social networks, media websites, and news portals. In industry, the two-tower model [1–6] is widely used in recommendation systems in large commercial Internet companies such as Microsoft [1], YouTube [2,3], Facebook [4], Baidu [5], and Alibaba [6]. Deep learning is an important study topic in the field of machine learning, and it has led to recent advancements in image processing, natural language understanding, speech recognition, online advertising,

and recommendation systems, among other areas. Integrating deep learning into recommendation systems and researching how to integrate massive multi-source heterogeneous data to build user models that better match user preferences and needs in order to enhance the performance and user satisfaction of recommendation systems has become the primary objective of deep learning-based recommendation systems. Typical recommender systems include those based on deep neural networks such as RNN [7], CNN [8], GCN [9], and GNN [10].

In recent years, the number of academic literature has increased tremendously, and discovering the required academic literature among many types of literature resource sites has become a crucial aspect of conducting scientific research. In the large amount of literature, it is challenging to locate valuable and actually needed literature that suits the criteria of the field. Currently, information retrieval and recommendation are the most common ways to obtain literature-related data. The information produced by the retrieval method does not reflect the features of the field, does not make use of domain knowledge, does not reflect the personalization of the retrieved information, does not make use of additional correlation information, and the accuracy should be improved. The information obtained by the recommendation system is more accurate, can accommodate more knowledge and information as input features for model computation, and can meet the demand for individualized recommendations. There are numerous approaches for acquiring information in which recommendation systems and information retrieval coexist and support one another. Methods for literature recommendation that incorporate information retrieval, domain knowledge, and other data are crucial for acquiring effective literature. Four groups of literature recommendation approaches can be distinguished: content-based filtering [11,12], collaborative filtering [13–16], graph-based recommendation algorithms [4,10–12,17–19], and hybrid recommendation algorithms [20–23].

Researchers have paid close attention to the digital resource system. When a search is performed, a large amount of academic literature that meets the search criteria is obtained. Still, not all of them correspond to the corresponding domain, which is indeed required.

In earlier studies, content-based filtering, collaborative filtering, and citation relationship-based methods were more prevalent. The content-based filtering approach [24] is simple. The researcher's publications or other information are cited first in the content-based filtering method to construct their profile. Then, the keyword similarity between the researcher's profile and the candidate literature is computed, and the candidate literature is rated accordingly. The basic idea behind the collaborative filtering method [25] is to identify the similarity between different users based on whether they have interaction behavior on the same items and then use similar users' preferences to recommend literature. However, each of these two approaches has the following drawbacks: (1) the content-based approach is a typical user-centric recommendation process. This type of approach is primarily based on historical information that the user has already viewed or manipulated. Using it only as a cue to recommend papers to users can face many challenges; (2) the effectiveness of collaborative filtering approaches is limited by the cold-start problem, and the results are often impersonal.

To alleviate the shortcomings of the two methods to a certain extent, further improve the performance of literature recommendation, and make more use of the characteristics of the literature itself, such as the correlation relationship between the cross-citations of the literature and the correlation relationship between the authors triggered by the literature, more and more researchers have introduced graph-based methods into literature recommendation. Graph-based approaches [26–30] mainly focus on the construction of a graph. The graph can be composed of citation networks, social networks, etc. Researchers and literature are different nodes on the graph. The relationships between researchers, between researchers and literature, and between literature and literature can be seen as edges between nodes.

There have been some promising research outcomes in the literature recommendation, but most existing systems suffer from the following issues. Current recommendation systems retrieve relevant literature after the researcher inputs keywords; however, the scope

of the literature is too broad, and the period is too long, resulting in a large amount of literature that may contain outdated research directions. In contrast, the researcher desires the quickest access to the current research hotspots in his field. In other words, most existing literature recommendation systems only consider keywords-related literature. They do not closely integrate with the current field hotspots, prioritizing the display of literature related to research hotspots because the recommended literature frontier is insufficient.

There have been some promising research outcomes in the literature recommendation, but most existing systems suffer from the following issues. Current recommendation systems retrieve relevant literature after the researcher inputs keywords; however, the scope of the literature is too broad, and the period is too long, resulting in a large amount of literature that may contain outdated research directions. In contrast, the researcher desires the quickest access to the current research hotspots in his field. In other words, most existing literature recommendation systems only consider keywords-related literature. They do not closely integrate with the current field hotspots. Secondly, with the exponential increase in the amount of literature, there is a continuous integration of relevant knowledge among various domains. However, when making recommendations, most existing literature recommendation systems do not adequately account for the relevance of domain knowledge, i.e., they do not effectively correlate with current domain knowledge and cannot fully utilize the correlation relationship between domain knowledge for the deep recommendation of literature. It prevents researchers from obtaining targeted literature in the relevant domain; i.e., it is not favorable to the transmission of information and does not attain certainty in literature recommendation. In the recommendation system, for instance, "neural network" has an entirely different meaning than "human body" when entered by researchers. Existing literature recommendation systems do not propose literature for domain-specific characteristics. As a result of this issue, researchers can obtain literature that is irrelevant to their research domain.

This research proposes a heterogeneous network literature recommendation method based on domain knowledge graphs and hotspot information composition to overcome the above-mentioned issues. In this paper, we argue that the knowledge network of domain knowledge involved in search information helps recommend more relevant literature and that the network of hotspot information involved in search information helps recommend literature related to current query domain hotspots.

Based on the authors' earlier work [31], the composition of the domain knowledge graph [31] has been accomplished. This paper extracts the relevant portion of the knowledge graph in the existing domain knowledge graph based on the query words and constructs the query-related domain knowledge triad. Similarly to the preliminary work I completed on the hotspot information network, this paper obtains the relevant portion of the hotspot sub-network according to the query term in the existing hotspot network, which constitutes the hotspot triad related to the query term, employs the triad fusion method to fuse the relevant two types of triads [32] into the corresponding query vector, and calculates the similarity between the query vector and the document vector in the candidate set, as shown in Figure 1.

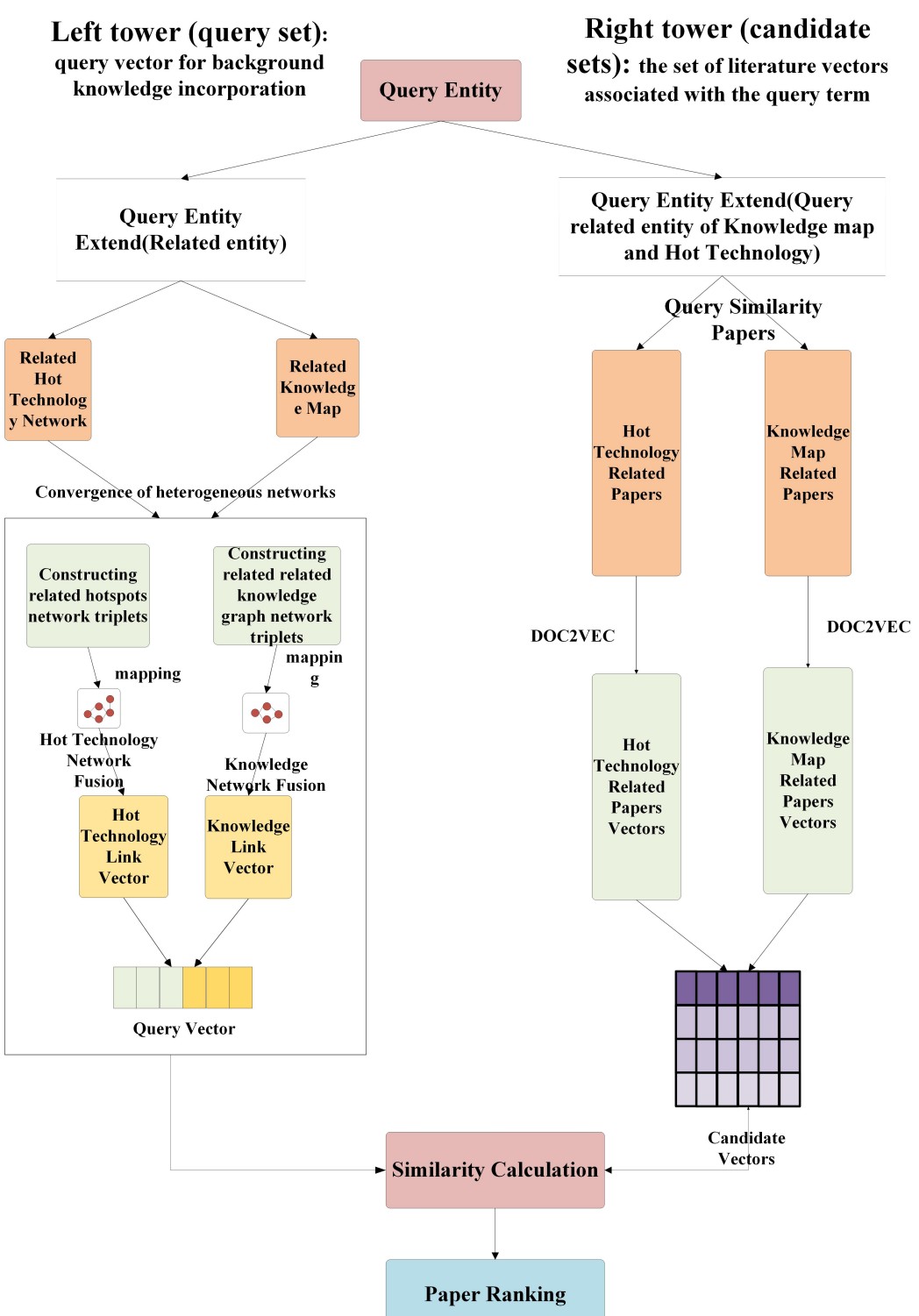

**Figure 1.** A noval scientific paper recommendation model via heterogeneous network embedding-based knowledge graph and hotspots.

The research objective of this paper is to provide a more accurate method for searching domain literature that reflects the characteristics and hot information of the domain given the query words or phrases. The main contributions of this paper are as follows.

- To provide a more accurate method for searching domain literature that reflects the characteristics and hot information of the domain. A heterogeneous network literature recommendation method is proposed based on the domain knowledge graph and

hotspot information composition. The combined effect of the domain knowledge graph and hotspot information is considered in the heterogeneous literature network for the first time. It overcomes the problem that the recommended literature needs more domain performance and reflects the hotspots and frontiers of literature. The effect of domain knowledge and hotspot information in literature recommendation is demonstrated. Experiments further validate the effectiveness of the method.

- The use of a proven two-tower retrieval system is appropriate. How to construct models that reflect domain characteristics and hotspot information under the two-tower system, thus enriching the background information of query terms, is a problem worthy of study. This paper proposed a triplet fusion method on a two-tower retrieval model. It is challenging to map the fusion of heterogeneous networks consisting of knowledge graphs and hotspot networks into the same hidden space. In this paper, based on the popular two-tower retrieval model in industry, which can ensure the basic performance of the model at a reasonable level, the knowledge graphs and hotspots related to query terms are fused into a unified query vector as a query set, which is ranked and recommended after similarity calculation with the candidate set.

- The search process was full of candidate documents. How to filter the enormous amount of candidate documents initially, select the appropriate set of candidates, and reduce the model's computation is a problem worth studying. This paper proposed a prospective literature extraction approach based on hotspots and knowledge graph information. By calculating the document similarity, the documents associated with the query word hotspot information or knowledge graph are extracted, reducing the number of irrelevant documents in the candidate document collection and increasing the matching efficiency.

- The experiment demonstrated the validity of our proposed model PRHN. We validated the new state-of-the-art method on a public dataset and our metallurgical domain literature. Experiments reveal that using heterogeneous networks comprised of knowledge graphs and hotspot networks enriches the query information and improves its precision while alleviating the issue of sparse data for some queries.

The next part of the article is organized as follows: In Section 2, related literature, we discuss four methods commonly used for literature recommendations. The recommendation models we use are described in Section 3. We detail our proposed citation recommendation model including the construction of the bibliographic network, heterogeneous entity embedding and the structure of our recommendation model. In the experimental section, we performed a validation comparison of two public datasets and one metallurgical dataset. Conclusions are presented in Section 5.

## 2. Related Work

### 2.1. Collaborative Filtering in Paper Recommendation

The CF model focuses on the actions or ratings of the literature by other users similar to the user. The model utilizes the user–paper interactions to generate recommendations and can result in strong performance [33,34]. In early applications, the collaborative filtering model achieved better results. Bansal [35] et al. used GRU in the paper collaborative filtering model for paper prediction. Sugiyama [36] et al. used the relationship between literature and citation for their collaborative filtering recommendation. McNee [37] et al. used the similarity relation between two attributes of user and item for their collaborative filtering recommendation. Wang [13] et al. built inter-topic correlations and provide an interpretable latent structure for users and items, which can form recommendations about both existing and newly published articles. Kang [38] et al. construct a Top-*n* recommendation using a user-item low-rank matrix to score papers. However, a common drawback of CF models is the cold start problem and insufficient feature information, which is severe in our academic recommendations when using real user-paper interaction data.

## 2.2. Content-Based Filtering Methods in Paper Recommendation

Content-based literature recommendation focuses on various types of information within the literature, such as textual features (item representation) and user profiles, which enrich the information about the characteristics of the research object. The content-based model generally learns the information within the literature for representation and finds similar papers after forming the corresponding feature vectors. Felice Ferrara [39] et al. introduce a content-based recommendation approach where a key-phrase extraction module is used to describe both papers' contents and user interests. Amami [40] et al. extract topics from their authored papers using LDA topic modeling to generate candidate article representations and then compute similarities between them to generate final recommendations. Achakulvisut [41] et al. use LSA to analyze the content of the literature and create recommendations based on similarity algorithms. Bhagavatula [42] et al. analyze the content of the literature and uses a neural network to map the content into a vector space for similarity ranking. The content-based analysis does not consider user engagement with the literature, has limitations in the content information studied and faces the problem of cold starts.

## 2.3. Graph-Based Methods in Paper Recommendation

There are citation linkages between documents, co-author affiliations, reader associations, knowledge background associations among literature, etc. These connections form a naturally heterogeneous network. It is more suited to convey the information in the literature collection via heterogeneous networks—traditional graph-based paper recommendation models [43,44] employed random walk methods to generate citation recommendations. Gori [43] et al. regarded authors, papers, topics, and keywords as nodes and their relationships as edges and proposed a citation relationship graph combined with a random walk for ranking literature by the PageRank method.

In a subsequent investigation, the random wandering network representation eliminates the cold start issue of the graph-based method. Manju [44] et al. propose a social network-based research paper recommendation method that alleviates cold start problems by incorporating users' social interaction using a novel approach to random walk Ergodic Markov Chain. The random wandering approach does not fully utilize the information in the heterogeneous network. Recent research has mapped the information in the heterogeneous network into a low-dimensional space for graph embedding [45–47] and network embedding [48–50]. Gupta [51] et al. used a graph embedding of reference composition with a vector representation of the content of the literature to recommend literature. Xiangjie Kong [20] et al. used a representation learning method combining citation network structure and paper content to obtain a vector of papers with a network structure and then used a similarity calculation method for recommendation to achieve better results. Liu [27] et al. propose a keyword-driven and popularity-aware paper recommendation approach based on an undirected paper citation graph. The graph-based recommendation considers various explicit and implicit correlations between documents and provides a more relevant approach to document recommendation.

## 2.4. Hybrid and Embedding-Based Methods in Paper Recommendation

Each of the methods listed above has different variations. The literature set possesses both heterogeneous network characteristics and content properties. The hybrid approach can make more extensive use of all types of information in the literature and effectively combine the above methods. The collaborative filtering approach does not consider the role of the literature's content, the content-based approach does not consider the association relationship between the literature, and the graph-based literature recommendation algorithm cannot function when the literature is not cited. The hybrid literature approach combines the benefits of various methods to improve recommendation efficiency and quality. Hammou [52] et al. combined collaborative filtering with content-based recommendations to

generate recommendation lists. XinYi L [53] et al. combined literature content with user click behavior data to show the advantages of hybrid data.

In recent years, with the development of graph embedding techniques and literature representation, more and more literature recommendations use a combination of content representation of literature and graph structure representation. Lee [54] et al. combine content-based and graph-based recommendations to alleviate the disadvantages of each approach. Kong [20] et al. use a combination of representing literature with word vectors and embedding information in a graph structure composed of literature citations to exceed the current effect of the literature recommendation baseline model.

Although the hybrid recommendation model outperformed the current model in some cases, such as when making recommendations in the literature with a solid domain, the characteristics of the domain need to be reflected, and the recommended literature is sometimes non-domain literature, particularly in metallurgical literature. Furthermore, the recommendation of previous literature does not consider the importance of the hotspot of literature for literature recommendation, and the recommended literature does not include all of the hotspot literature in the current field. In contrast, for literature, the recommendation of hotspot literature in the area is essential to grasp the field's frontier. The method proposed in this paper aims to address the issue that the recommended literature lacks domain characteristics and hotspot information. The four methods are summarized in Table 1.

**Table 1.** Advantages and disadvantages of different methods in literature recommendation

| Method | Advantages | Disadvantages |
|---|---|---|
| Content-based filtering methods in paper recommendation | Simple, using various parts of the paper contents. | A typical user-centric recommendation process. This type of approach is primarily based on historical information that the user has already viewed or manipulated. |
| Collaborative Filtering in paper recommendation | Utilize the user–paper interactions to generate. | The cold-start problem and the results are often impersonal. |
| Graph-based methods in paper recommendation | This method considers the influence of various correlations between and outside the literature on the recommendation results. | The recommendation results are unsatisfactory if the nodes have only a few associational relations. The graph data pre-processing workload is large, and the model is inefficient in processing graph data. |
| Hybrid and embedding-based methods in paper recommendation | Can make more extensive use of all types of information in the literature and effectively combine the above methods. | The recommended literature is sometimes non-domain literature; Integration of various methods sometimes leads to inefficiencies and worse results. |

## 3. Methodology

The previous recommended literature does not address the problem of reflecting domain hotspots and the problem of cold start in the literature recommendation process to address the inaccuracy of literature recommendation in the domain. In this paper, we propose a vector-matching recommendation method for literature recommendation that fuses domain knowledge graphs and domain hotspots and outperforms the current baseline model.

The paper firstly relies on the authors' previous research to achieve the following: extract the domain knowledge subgraphs related to query terms from the existing domain knowledge map; construct the domain hotspot subgraphs related to query terms from the

current domain literature hotspot identification model; extend the query terms to make the query content rich in domain knowledge and domain hotspot information, transform the heterogeneous network formed by the domain knowledge and domain hotspot from the query term extension to the relevant model and then carry out vectorized representation, i.e., vectorized representation of query terms, calculate the similarity between the query term vector and the preprocessed literature vector set and then turn it into a TOP-*N* problem for solving; see Figure 2. For example, when we use "intelligent metallurgy" as the search query term, based on the domain knowledge map and hot words set formed by the author's previous research, the domain knowledge subgraphs with "heat and mass transfer, metallurgical kinetics, pyrometallurgy, hydrometallurgy, phase change" as the backbone were formed. At the same time, we have created the subgraphs of "metallurgical first-principles calculation, mineral genetics, big metallurgical data, molecular mechanism, and special field metallurgy" as the backbone hot. The domain knowledge subgraph and the domain hotspot subgraph are fused into a query vector by the model algorithm, and the most suitable relevant literature is recommended after matching with the filtered candidate literature set; see Figure 2.

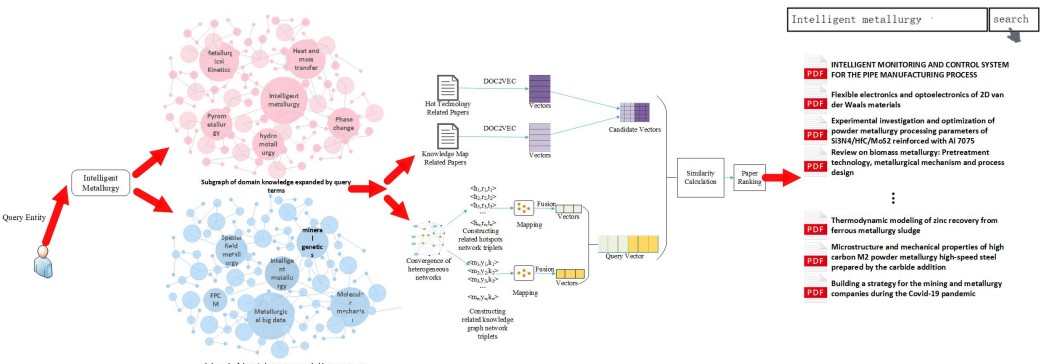

**Figure 2.** Example of literature recommendation model

### 3.1. Problem Formulation

Input: For query terms *Query* abbreviated as *Q*.

Output: Let the set of the domain papers be *P*. Find the documents from the collection of articles whose first *n* items are closest to *Q*.

Because the query term *Query* contains limited information and does not fully reflect domain characteristics and domain hotspot information, we extend the domain knowledge and extract the corresponding hotspot information in this paper to form a domain knowledge graph and a domain hotspot graph for the query term *Query*. The mapping information is used to construct the query vector for calculating similarity with the literature set.

We clustered the entities in the knowledge graph that the authors' previous research created. The entities in the categories similar to the query term *Query* and the relationships between the entities served as the query term's extended domain knowledge.

We clustered the hotspot entities formed by the authors' previous research. The entities similar to the query term *Query* and their relationships were used as the query term's extended hotspot context.

By query terms *Query*, extended domain knowledge, and extended hotspot background, our model *f* recommends the top *n* items in the candidate set of documents $p_n$.

$$\tilde{p}_n = f(Query, KnowledgeGraph, HotspotGraph, PaperSet, \theta) \tag{1}$$

where $\tilde{p}_n$ is the top *n* items predicted in the literature recommendation algorithm, *f* is the literature recommendation model method, *Query* is the query term, *KnowledgeGraph* is the domain knowledge graph constructed based on query term *Query* expansion,

*HotspotGraph* is the hotspot graph constructed based on query term *Query* expansion, and *PaperSet* is the set of query candidates for query term *Query*.

The objectives of the model are:

$$\arg\min_{\theta} \sum_{q \in Query} \sum_{j \in paper_j} p_{q,j} - \tilde{p}_{q,j} \tag{2}$$

where $p_{q,j}$ is the labeling score of query term *Query* for candidate $j$ and $\tilde{p}_{q,j}$ is the prediction score of query term *Query* for candidate $j$.

### 3.2. Construction of Knowledge Graphs and Domain Hotspot Information Networks Related to Query Terms Query

The effective extension of query terms *Query* to introduce domain knowledge and hotspot information related to query terms *Query* as input information into the model is the highlight of this paper. It is a challenging problem to establish rich and effective domain knowledge and hotspot information. The domain knowledge graph, which corresponds to domain knowledge related to query terms *Query*, and the hotspot relationship graph, composed of hotspots mined from the literature by the author's previous research, are fused into the graph model in this paper. The model can reflect the domain characteristics and hotspots by incorporating the knowledge graph and the hotspots information network.

$$KnowledgeGraph = \{(h, r, t) | h, t \in Entitys, r \in R\} \tag{3}$$

where *Entitys* represent the set of entities associated with the query vector, and relations represent the set of relations. Cosine similarity is used to compute similarity in this case. We compute similarity for the entities in the domain knowledge graph constructed by the author's previous work [31]. Using $CosineSimilarity = CosineSimilar(query, entity)$ and $ConsineSimilarity > 0.5$, we consider the domain entity to be related to the query term; the entity and the entity association relationship formed on the graph and the entity at the other end are saved, i.e., the corresponding $(h, r, t)$ is saved. In order to go further to find more similar entity relationships, with reference to Haoyu Wang [55], a specific gravity threshold $\lambda$ was experimentally proposed, and the triad of entities above the threshold was saved and other entities were removed.

$$\lambda = \frac{\exp(CosineSimilar(query, entity_i)/T)}{\sum_i \exp(CosineSimilar(query, entity_i)/T)} \tag{4}$$

where $T$ is the adjustment parameter, the larger the parameter, the less critical the similarity comparison is. After the censoring of this threshold, more similar entities and entity relationships are kept.

*HotspotGraph* is a triplet of domain hotspots associated with the query term *Query*, i.e.,

$$HotspotGraph = \{(h, r, t) \mid h, t \in HotEntitys, r \in R\} \tag{5}$$

Similarly, the construction of hotspot information networks is similar to the construction of domain knowledge graphs and will not be repeated.

### 3.3. Representation of Query Terms in the Left Tower

The vector of query sets $e_{query}$ consists of a vector of related knowledge graphs $e_{KnowledgeGraph}$ and a vector of corresponding hotspot graphs $e_{HotspotGraph}$ stitched together.

$$e_{query} = e_{KnowledgeGraph} \| e_{HotspotGraph} \tag{6}$$

$$e_{KnowledgeGraph} = f_{GCN} = LeakRelu(w(e_{Entity} + e_{net} + e_{LightKG})) \tag{7}$$

The knowledge graph vector $e_{KnowledgeGraph}$ associated with the query vector is formed by the vector of words $e_{Entity}$ corresponding to the query term *query*, the vector $e_{net}$ formed

by the fusion of the knowledge graph triad after the similarity calculation in the knowledge graph, the vector $e_{LightKG}$ formed by the fusion of the most relevant terms or words identified by the query term *query* using the method of *LightKG* [55], and the vector formed by the summation of the three vectors and the multiplication with the trainable weight matrix coefficients $w$ after the GCN feature extraction.

$$e_{net} = \sum_{(h,r,t) \in net} \pi(h,r,t)e_t \tag{8}$$

where $\pi(h,r,t)$ is the propagation factor, which indicates how much head information $h$ is propagated to tail information $t$ through the relationship $r$ for each involved knowledge graph triad.

$$\pi(h,r,t) = (w_r e_t)^{\mathrm{T}} \tanh(w_r e_h + e_r) \tag{9}$$

where $w_r \in R^{k \times d}$ is the transformation matrix of relation $r$, which projects entities from the $d$-dimension entity space into the $k$ dimension related space. In order to facilitate the model calculation, $\pi(h,r,t)$ is normalized in this paper.

$$\pi(h,r,t) = \frac{\exp(\pi(h,r,t))}{\sum_{(h,r',t') \in net} \exp(\pi(h,r',t'))} \tag{10}$$

where $e_t$ is the embedding information corresponding to the tail information $t$ in the triad $(h,r,t)$ trained by the loss function $L$ in the triad embedding training layer, $e_h$ is the embedding information of the head information $h$ in the triad $(h,r,t)$, and $e_r$ is the embedding information of the relationship information $r$ in the triad $(h,r,t)$, such that $e_h + e_r \approx e_t$, and the corresponding loss function in the training process is:

$$L_{KG} = \sum_{(h,r,t,t') \in T} -\ln \sigma(f(h,r,t') - f(h,r,t)) \tag{11}$$

where $T = \{(h,r,t,t')|(h,r,t) \in G, (h,r,t') \notin G\}$, $(h,r,t')$ is a broken triplet constructed by replacing one entity in a valid triplet randomly. $\sigma(\cdot)$ is the sigmoid function. $f(h,r,t)$ is the judgment scoring function of formula $e_h + e_r \approx e_t$: the lower the score, the more it meets the requirements of the formula.

$$f(h,r,t) = \|w_r e_h + e_r - w_r e_t\|_2^2 \tag{12}$$

Similar analogy:

$$e_{HotspotGraph} = f_{GCN} = LeakRelu(w(e_{Entity} + e_{LightKG})) \tag{13}$$

Considering that the relationships in the triples in the domain hotspot network are relatively homogeneous, the $e_{net}$ is removed to simplify the operations and improve the efficiency of the processes, leaving only the $e_{LightKG}$ [55]. The vector $e_{LightKG}$ contains information about the top relevant hotspots in the hotspot network that are most relevant to the query term. Other relevant sub-formulas are analogous.

### 3.4. Construction of the Right Tower Candidate Set Literature Vector Matrix

In this paper, based on the mature query retrieval left and right-tower models in the industry, we extend the knowledge of query terms and hotspot background information while using semantic similarity on the right-tower model for the initial screening of the literature. Finally, the retrieval problem is converted into a recommendation problem for solving, and good results are achieved. In the query term expansion stage, this paper uses the expansion term method in 3.2 to expand the query terms into relevant knowledge graphs and hotspot terms. Then, the traditional search paper interface is invoked to search the query terms, the expanded knowledge mapping terms, and the related hotspot terms one by one, and the top 100 papers retrieved for each term are put into the corresponding



literature database to form a candidate set. After that, the Doc2vec method is called to convert the candidate set into a vector for storage, as in Figure 1.

### 3.5. Model Prediction and Optimization

In this paper, the query vector $e_{query}$ formed by query terms and extended background information is regarded as the user information $e_{user}$ in the recommendation system, i.e., $e_{user} = e_{query} = e_{KnowledgeGraph} \| e_{HotspotGraph}$; then, the query vector matrix $Query_{user} = (e_{user_1}, e_{user_2}, \ldots, e_{user_i}) = (e_{query_1}, e_{query_2}, \ldots, e_{query_i})$.

Each literature vector $e_{item} = e_{paper}$ in the candidate set; then, the candidate set vector matrix $Papers_{item} = (e_{paper_1}, e_{paper_2}, e_{paper_3}, \ldots, e_{paper_i})$. $\tilde{p}_{u,i}$ is the prediction score of query term $Query$ in the model for candidate $i$. Then:

$$\tilde{y}(u, i) = \tilde{p}_{u,i} = Query_{user}^{\mathrm{T}} * Papers_{item} \tag{14}$$

The optimization of the PRHN model uses a combination of the domain knowledge graph loss function $L_{KG}$, the hotspot network loss function $L_{Hotspot}$ (analogous to $L_{KG}$ and will not be repeated), and the scoring function $L_{Score}$.

$$L = L_{KG} + L_{Hotspot} + L_{Score} \tag{15}$$

$$L_{Score} = \sum_{u \in query} \sum_{i \in paper} (y(u, i) - \tilde{y}(u, i))^2 \tag{16}$$

The scoring function $L_{SCORE}$ is the summation of the squared difference between the token score and the prediction score of the query term $Query$ for candidate document $i$ in the model.

## 4. Experiments and Results

### 4.1. Datasets

We used the base datasets AMiner, DBLP, and our own collected metallurgical dataset to validate the model.

The AMiner (https://www.aminer.cn/data/?nav=openData accessed on 1 February 2022) dataset contains 500,000 relevant computing papers (each includes metadata such as title, author, abstract, publication organization, and year), 10,000 knowledge concepts, concept relationships, concept definitions, 200,000 expert information, and 9668 keywords. The results of the various methods on the general data set are compared with the results of each method on the metallurgical data set, and it is proposed that the methodology proposed in this paper is more effective on the metallurgical data set.

The DBLP (https://dblp.uni-trier.de/xml/ accessed on 23 February 2022) dataset is a computer science literature database. Its subset includes 14,475 authors, 14,376 literature, and 8920 phrases, as well as 41,794 author literature relations and 114,624 literature term relations. The research fields covered in the literature include databases, data mining, artificial intelligence, and information retrieval. We manually scored and labeled the phrases (terms) and ran experimental comparisons on different models.

The metallurgical dataset is a Chinese literature dataset in the field of metallurgy that was created by crawling the Chinese metallurgy literature from the China Knowledge Network and pre-processing the data in this paper. It includes 3268 authors, 5702 literature, 2000 phrases, 7375 author–literature relationships, 37316 literature–term relationships, and so on. During the pre-processing stage, I will use previous research to build a knowledge graph of the metallurgical domain for the literature, using metallurgical domain knowledge terminology triad formation. The method of pre-processing study is used to perform hotspot mining of the literature to form a triad of hotspot terms or phrases in the field of metallurgy. As inputs to the model, we use query terms and queries to extend background knowledge and hotspot networks. We divided the above dataset into three parts: 60% for training, 20% for testing, and 20% for validation. K-fold cross-validation is

used to evaluate the effect of the model and select the best super-parameter. The test data were divided into multiple batches for verification.

### 4.2. Baselines

ML-DTR Model [35]. The ML-DTR Model encodes the articles' text sequence into a vector representation employing GRUs. To be consistent with the ML-DTR, the content of articles is used to implement the model. This method uses the title and content of the paper as the input data for the model.

LightGCN [9]: Graph convolution has become a widespread technique in recommender systems for collaborative filtering methods. Feature transformation and nonlinear activation have little effect on collaborative filtering performance. As a representative model for collaborative filtering recommendation, lightGCN discards the two standard GCN operations of feature transformation and nonlinear activation: simple, linear, concise, easier to train, better generalization ability, and representative. Because GCN cannot apply relationship information between entities, this paper uses query terms and extensions as LightGCN input data.

VOPRec [20]: VOPRec is a state-of-the-art hybrid paper recommendation model (GB+CB) in which the core idea is to represent papers with vectors in the citation network so that the similarity between papers can be calculated using the vectors. Specifically, it uses word2vec and structural2vec to reconstruct a citation network to derive representative vectors through graph learning and recommends similar papers to scholars according to their cosine similarity. In this paper, the entities of query terms and query extension terms have been used as the citation information of the paper as the input data of VOPRec.

PRHN-without KG (paper recommendation model via heterogeneous network embedding without knowledge graph): The ablation model with the knowledge graph was removed from the proposed two-tower recommendation model. The query terms and the hotspot network extended by the query terms serve as the model's input data.

PRHN-without hotspot (paper recommendation model via heterogeneous network embedding without hotspot): This paper proposes an ablation model for removing domain hotspot information in a two-tower recommendation model. The query terms and the domain knowledge graph extended by the query terms are used as the model's input data.

PRHN (paper recommendation model via heterogeneous network embedding): This paper presents a proposed two-tower recommendation model combining a domain knowledge graph and hotspot information. The query terms, domain knowledge graph, and hotspot network, extended by the query terms, are used as the model's input data.

### 4.2.1. Evaluation Methodology

All experiments are run on the same machine (Intel Xeon 8-Core CPU of 2.4 GHz and single NVIDIA GeForce GTX TITAN X GPU) for fair comparison. We apply the widely used leave-one-out technique (Gao et al. 2019; Rendle et al. 2009; Chen et al. 2020d) and then adopt two popular metrics, HR (Hit Ratio) and NDCG (Normalized Discounted Cumulative Gain), to judge the performance of the ranking list.

HR (Hit Ratio) is the ratio of the total number of documents to the number of documents in the dataset that appear in the Top-$N$ recommendation list. The range of HR-value is 0–100%.

$$\text{HR} = \frac{N_{\text{top}}}{\text{Number}_{\text{total}}} \tag{17}$$

The NDCG (normalized discounted cumulative gain) score of a recommendation list $L$ can be derived by:

$$\text{NDCG}_L = \frac{\text{DCG}_L}{\text{IDCG}_L} \tag{18}$$

where $DCG_L$ can be formulated as:

$$DCG_L = \sum_{i=1}^{n} \frac{2^{\text{rel}(i)} - 1}{\log_2(i+1)} \tag{19}$$

where the $\text{rel}(i)$ represents the rank value for item $i$ in the true list and $IDCG_L$ is the DCG value of the ideal order of the cited papers in our case. The range of NDCG-value is 0–1.

For each user, our evaluation protocol ranks all the items except the positive ones in the training set. In this way, the obtained results are more persuasive than ranking a random subset of negative items only [56]. For each method, we randomly initialize the model and run it five times. After that, we report the average results.

### 4.2.2. Parameter Settings

The parameters for all baseline methods are initialized as in the corresponding papers, and they are then carefully tuned to achieve optimal performances.

After the tuning process, the batch size is set to 20 in ML-DTR, 50 in others. The latent dimension of all the model dimensions is set to 256 in all models. The learning rate is set to 0.001. To prevent overfitting, the dropout ratio is set to 0.5 in LightGCN and 0.1 in others.

Our experiments are conducted with PyTorch running on GPU machines (NVIDIA Tesla V100).

We selected typical models related to graph convolution for comparison on DBLP, AMiner, and Metallurgical datasets, and the results are presented in Tables 2–4, as shown in Figures 3–5. The recommended lengths $K = 10, 20, 50$ were chosen for comparison in the experiments to verify the output effect of different recommendation lengths, resulting in the following observations:

**Table 2.** Performance of different models on Aminer datasets.

| Aminer Datasets | HR@10 | HR@20 | HR@50 | NDCG@10 | NDCG@20 | NDCG@50 |
|---|---|---|---|---|---|---|
| ML-DTR | 0.045 | 0.148 | 0.66 | 0.013 | 0.03 | 0.14 |
| LightGCN | 0.048 | 0.149 | 0.665 | 0.0135 | 0.037 | 0.145 |
| VOPRec | 0.049 | 0.15 | 0.669 | 0.0139 | 0.038 | 0.15 |
| PRHN without hotspot | 0.049 | 0.156 | 0.666 | 0.016 | 0.041 | 0.151 |
| PRHN without KG | 0.051 | 0.154 | 0.679 | 0.019 | 0.04 | 0.154 |
| PRHN | 0.051 | 0.205 | 0.705 | 0.015 | 0.064 | 0.168 |

**Table 3.** Performance of different models on DBLP datasets.

| DBLP Datasets | HR@10 | HR@20 | HR@50 | NDCG@10 | NDCG@20 | NDCG@50 |
|---|---|---|---|---|---|---|
| ML-DTR | 0.017 | 0.16 | 0.6 | 0.0059 | 0.04 | 0.13 |
| LightGCN | 0.018 | 0.165 | 0.61 | 0.006 | 0.041 | 0.133 |
| VOPRec | 0.019 | 0.168 | 0.62 | 0.0061 | 0.042 | 0.134 |
| PRHN without hotspot | 0.018 | 0.163 | 0.633 | 0.006 | 0.041 | 0.132 |
| PRHN without KG | 0.021 | 0.17 | 0.642 | 0.006 | 0.044 | 0.136 |
| PRHN | 0.025 | 0.179 | 0.66 | 0.007 | 0.045 | 0.139 |

**Table 4.** Performance of different models on metallurgical datasets.

| Metallurgical Datasets | HR@10 | HR@20 | HR@50 | NDCG@10 | NDCG@20 | NDCG@50 |
|---|---|---|---|---|---|---|
| ML-DTR | 0.39 | 0.4 | 0.6 | 0.26 | 0.28 | 0.31 |
| LightGCN | 0.4 | 0.45 | 0.65 | 0.27 | 0.29 | 0.34 |
| VOPRec | 0.43 | 0.49 | 0.68 | 0.29 | 0.31 | 0.35 |
| PRHN without hotspot | 0.418 | 0.539 | 0.701 | 0.309 | 0.364 | 0.374 |
| PRHN without KG | 0.435 | 0.545 | 0.709 | 0.316 | 0.354 | 0.381 |
| PRHN | 0.473 | 0.582 | 0.782 | 0.339 | 0.356 | 0.392 |

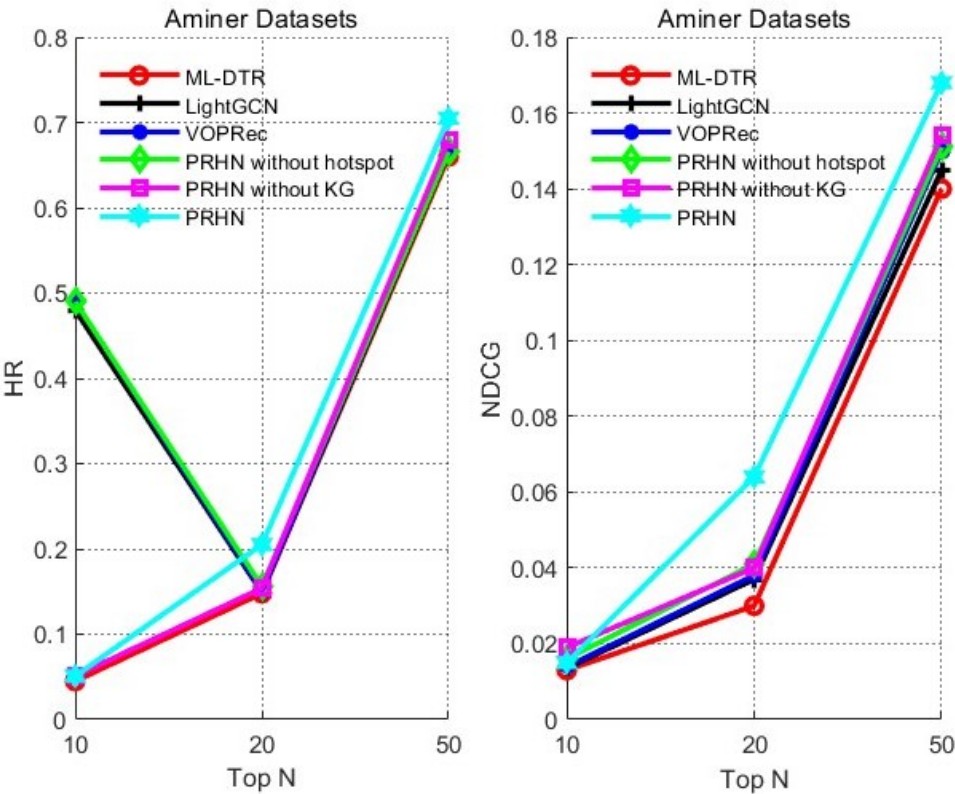

**Figure 3.** Performance of different models on Aminer datasets.

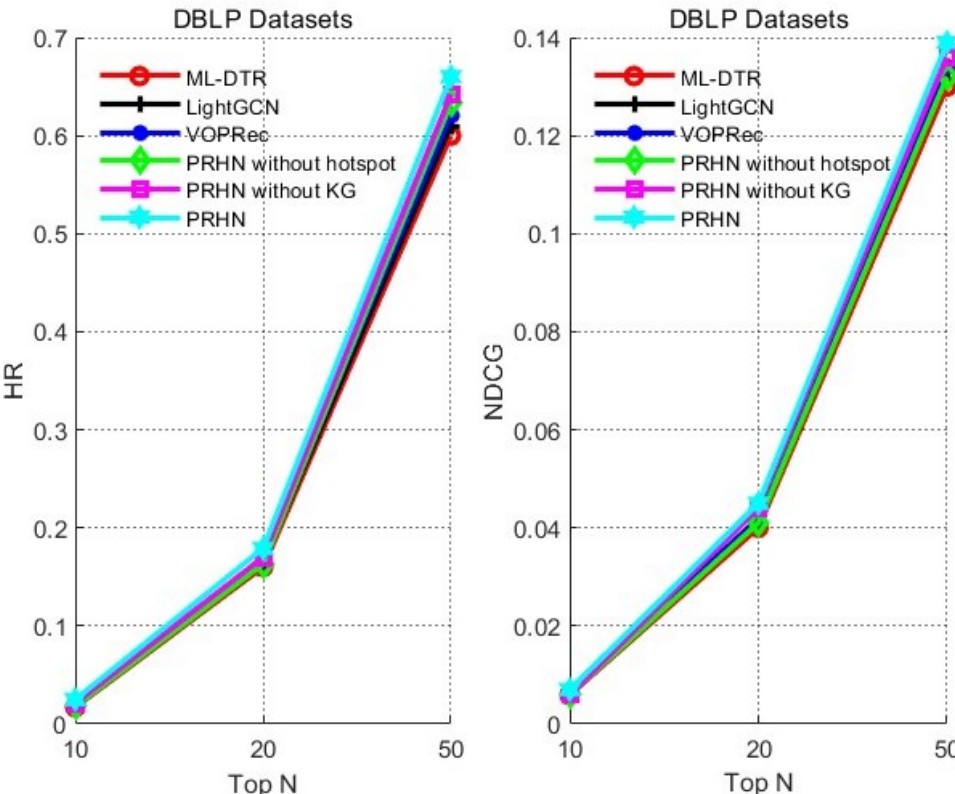

**Figure 4.** Performance of different models on DBLP datasets

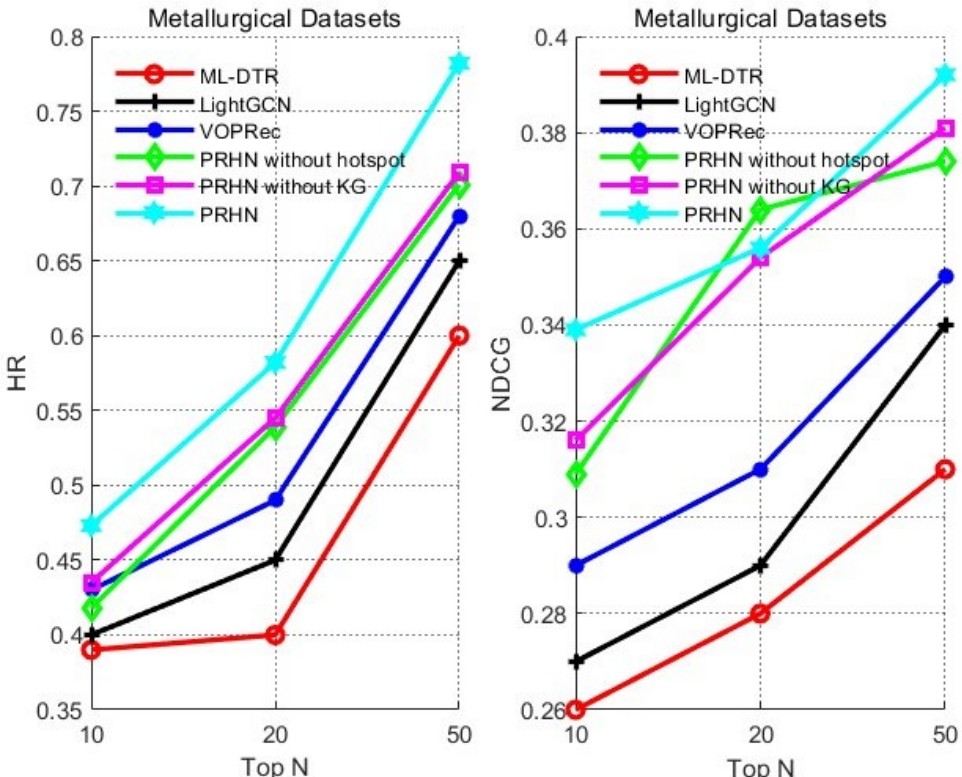

**Figure 5.** Performance of different models on metallurgical datasets

First and foremost, our proposed PRHN achieves the best performance on the three datasets, especially on the metallurgical datasets, which significantly outperformed all the state-of-the-art baseline methods with HR values and NDCG values higher than the others.

The main reason is our model, which incorporates domain knowledge into the computational model, especially in the field of metallurgy, where the influence of domain characteristics is more evident. On the other hand, the hotspot information embedded in the literature also has a guiding role in literature recommendation. Integrating hotspot network information into the two-tower recommendation model by aggregation further improves the model's effectiveness.

Second, the model proposed in this paper works better than the other models. It also illustrates again that literature recommendation should focus on the information inside and outside the literature, i.e., the correlation relationship between the literature. The correlation between the literature has a more significant effect on the literature recommendation.

Third, the model proposed in this paper is evaluated on various datasets. The best results are obtained on the metallurgical dataset, which again shows that the model is more appropriate for processing metallurgical data. The correctness and feasibility of the model selection method are also shown.

4.2.3. Ablation Experiments

In order to verify the modules, this paper decomposes the model for ablation experiments. The model is decomposed into a method PRHN-without KG that incorporates hotspot network information, a PRHN-without hotspot that contains only domain knowledge graph information in the model, and the PRHN model that includes both hotspot information and a metallurgical knowledge graph. We validated the experiments on three different datasets, and all of them yielded valid results, indicating that the approach used in this paper is practical. We obtained the best results when the top $K = 50$; Table 5 shows this as an example.

**Table 5.** Performance of different models on metallurgical datasets.

| | Aminer Datasets | | DBLP Datasets | | Metallurgical Datasets | |
|---|---|---|---|---|---|---|
| | **HR@50** | **NDCG@50** | **HR@50** | **NDCG@50** | **HR@50** | **NDCG@50** |
| PRHN without hotspot | 0.666 | 0.151 | 0.633 | 0.132 | 0.701 | 0.374 |
| PRHN without KG | 0.679 | 0.154 | 0.642 | 0.136 | 0.709 | 0.381 |
| PRHN | 0.705 | 0.168 | 0.66 | 0.139 | 0.782 | 0.392 |

The PRHN-without KG model is improved after adding hotspot information compared to other models, indicating that the method of adding hotspot information is compelling. The PRHN-without-hotspot model is also improved by including a metallurgical knowledge map, and it is more effective than the PRHN-without-KG model. The experimental results show that domain knowledge map information is more critical in literature recommendations than hotspot information. The PRHN has the most significant impact by including both hotspot information and a metallurgical knowledge map, verifying that the two types of information added are correct. The effectiveness of the two types of information added is also validated.

The method performs best on the metallurgical dataset, demonstrating that the method captures the domain dataset's characteristics and is targeted.

### 4.2.4. Correlation between the Size of the Recommendation List and the Results

As shown in Figures 6 and 7, it is clear from the results that our proposed model outperforms the other models in the paper recommendation list. While the size of the recommendation list increases, the NDCG-value and HR-value of our model increases. It can be observed that among the chain table lengths of 10, 20 and 50, the chain table length of 50 gives the best results.

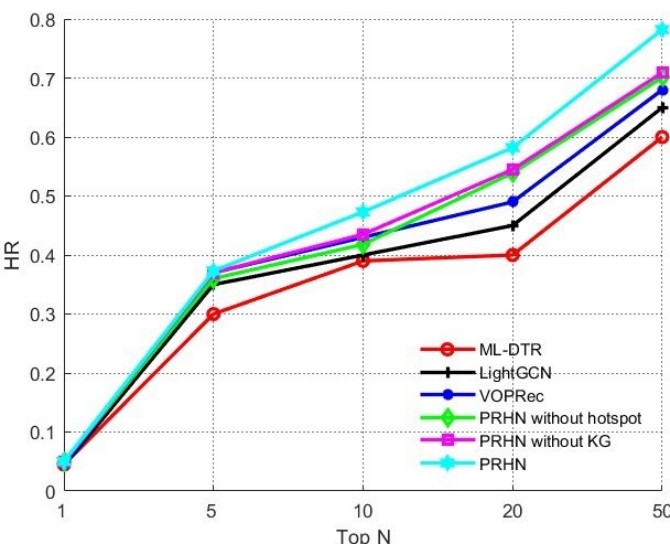

**Figure 6.** Performance metrics of HR for the three baseline models and our proposed model in metallurgical datasets.

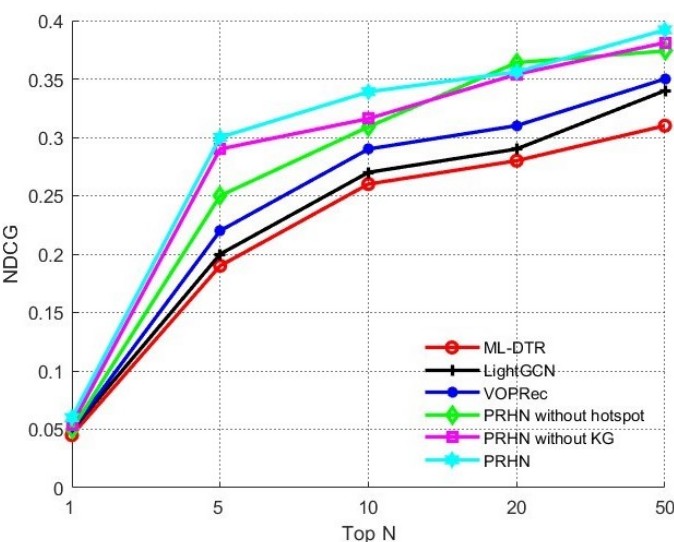

**Figure 7.** Performance metrics of NDCG for the three baseline models and our proposed model in metallurgical datasets.

## 5. Conclusions

Experiments show that hotspot information and domain knowledge significantly impact document recommendation. The heterogeneous literature recommendation method proposed in this paper employs a heterogeneous network to depict the literature's external and internal association relationships. It aggregates the hotspot information network and domain knowledge information network related to query terms into the heterogeneous network, making the literature recommendation method domain-specific and emphasizing the characteristics of the literature. Compared with existing models, our model has improved HR and NDCG by approximately 4.5% and 4.2%, respectively.

Previous models have yet to consider how to reflect the domain characteristics and how to depict the various types of associational relations among documents. The experiments in this paper show that heterogeneous networks of literature are essential for describing multiple types of relationships among documents. It is worthwhile to study how the heterogeneous network of literature can be combined with traditional models and how to make the various valuable information for literature recommendation effectively.

In addition, the model provides some relief for data scarcity and cold start problems. This method expanded the query terms so that uncommon query terms that may not initially be queried for the right paper can be queried after the information is extended.

Based on the achievements as well as limitations, we believe several directions still need to be further investigated. (1) The best results of the experiment appear when TOP $N = 50$, but when $N$ is relatively large, there are too many candidates for the recommended literature, making the recommendation unfeasible. Consider how to improve the effectiveness of recommendations when the list is short. (2) In addition to incorporating domain knowledge and hotspot information, constructing a heterogeneous network is also a worthy research problem to improve the recommendation effect by reasonably incorporating more related information.

**Author Contributions:** Conceptualization, W.C.; methodology, W.C.; software, W.C.; validation, Y.X., Y.W.; formal analysis, W.C.; investigation, W.C.; resources,W.C.; data curation, Y.Z.; writing—original draft preparation, W.C.; writing—review and editing, Y.Z., Y.X. and Y.W.; visualization, Y.Z.; supervision, W.C.; project administration, W.C.; funding acquisition, W.C. and Y.X. All authors have read and agreed to the published version of the manuscript.

**Funding:** This work is supported by Yunnan Basic Research Project (Grant Nos. 202001AT0 70046) and the Open Fund of Yunnan Key Laboratory of computer technology application in China (Grant No.140520200151).

**Institutional Review Board Statement:** Not applicable.

**Informed Consent Statement:** Not applicable.

**Data Availability Statement:** The data used to support the findings of this study are included within the article.

**Conflicts of Interest:** The authors declare no conflict of interest.

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
