# Peer review of "Hotspot Information Network and Domain Knowledge Graph Aggregation in Heterogeneous Network for Literature Recommendation"

_applsci, doi:10.3390/app13021093_

Round 1

Reviewer 1 Report

 1. Kindly Improve abstract sections and some impact full discussion.

2. In the end of introduction section add research objective, Motivation with contribution should be added. also break large paragraph into small paragraph.

3. In litrature survey, add comparison table.

4. If possible add some good some more graphical comparison in the result section.

5. proof read required

6. Conclusions having very less information.

Reviewer 2 Report

In this article, heterogeneous network literature recommendation method based on domain knowledge graph and hotspot information are used in an interesting topic. However, the manuscript contains several issues that requires clarification:

1. The Introduction part is too long. Author may consider to simplify the introduction part. What is PRHN actually stands for? The author did not explain and define the abbreviation/term used in the article. This also includes the term for LDA and LSA.

2. How does PRHN address the reflecting domain hotspots and cold start in literature recommendation problem? Please highlight at which section the author mentioned about this?

3. Any particular reason for having a bold sentences for last paragraph (pg 2) and 1st paragrapgh (pg 3)?

4. There is a repetition of "related" word in Figure 1 (Convergence of Heterogeneous Networks part)

4. The quality of Figure 2 is low and not readable.

5. Typo error in Section 2. (Related Work).

6. Please revise the title for Subsection 3.3. It seems like a sentence, not a sub-section title.

7. Please add description about the source of AMiner and DBLP dataset in the datasets section.

8. Is there any results to show obtained from the proposed PRHN model? Ie: the similarity score or the paper ranking?

9. Give more details about the Hit Ratio and NDCG. It is suggested to provide the formula used for both performance metrics and its range value.

10. Please refer each Table 1, 2 and 3 followed by explanation of the results for each table.
